# Effect of cis- and trans-Monounsaturated Fatty Acids on Palmitate Toxicity and on Palmitate-induced Accumulation of Ceramides and Diglycerides

**DOI:** 10.3390/ijms21072626

**Published:** 2020-04-09

**Authors:** Farkas Sarnyai, Anna Somogyi, Zsófia Gór-Nagy, Veronika Zámbó, Péter Szelényi, Judit Mátyási, Laura Simon-Szabó, Éva Kereszturi, Blanka Tóth, Miklós Csala

**Affiliations:** 1Department of Medical Chemistry, Molecular Biology and Pathobiochemistry, Semmelweis University, H-1085 Budapest, Hungary; 2Department of Inorganic and Analytical Chemistry, Budapest University of Technology and Economics, H-1111 Budapest, Hungary

**Keywords:** trans fatty acid, lipotoxicity, diabetes, endoplasmic reticulum stress, ceramide, diacylglycerol

## Abstract

Dietary trans fatty acids (TFAs) have been implicated in serious health risks, yet little is known about their cellular effects and metabolism. We aim to undertake an in vitro comparison of two representative TFAs (elaidate and vaccenate) to the best-characterized endogenous cis-unsaturated FA (oleate). The present study addresses the possible protective action of TFAs on palmitate-treated RINm5F insulinoma cells with special regards to apoptosis, endoplasmic reticulum stress and the underlying ceramide and diglyceride (DG) accumulation. Both TFAs significantly improved cell viability and reduced apoptosis in palmitate-treated cells. They mildly attenuated palmitate-induced XBP-1 mRNA cleavage and phosphorylation of eukaryotic initiation factor 2α (eIF2α) and stress-activated protein kinase (SAPK)/c-Jun N-terminal kinase (JNK), but they were markedly less potent than oleate. Accordingly, all the three unsaturated FAs markedly reduced cellular palmitate incorporation and prevented harmful ceramide and DG accumulation. However, more elaidate or vaccenate than oleate was inserted into ceramides and DGs. Our results revealed a protective effect of TFAs in short-term palmitate toxicity, yet they also provide important in vitro evidence and even a potential mechanism for unfavorable long-term health effects of TFAs compared to oleate.

## 1. Introduction

The cell damage caused by sustained fatty acid (FA) oversupply is referred to as lipotoxicity, and it is known to play a central role in the development of certain obesity-related metabolic disorders, such as non-alcoholic fatty liver disease (NAFLD) [1,2], insulin resistance [3] and type 2 diabetes mellitus [4,5]. Lipotoxicity induced in pancreatic β-cells is of particular importance as it hinders the compensation of insulin resistance, which intensifies the primary damage through further increase in FA release from the adipocytes. Several in vitro studies have been performed to investigate the molecular mechanism of β-cell lipotoxicity by using various insulinoma cell lines [6,7,8] and the major endogenous saturated and monounsaturated FAs, i.e., palmitate (16:0) and oleate (18:1 cis-Δ9), respectively. It has been demonstrated that the enhancement of apoptosis and autophagy that leads to a reduction in β-cell mass is largely due to a destructive endoplasmic reticulum (ER) stress [7,9], which, in turn, is aggravated by the accumulation of toxic ceramide intermediates [10,11]. Lipotoxic ER stress triggers signaling pathways that lead to pro-apoptotic transcriptional changes via X box-binding protein-1 (XBP-1) mRNA cleavage and phosphorylation of eukaryotic initiation factor 2α (eIF2α). In addition, severe malfunction of the ER leads to the activation of stress-activated protein kinase (SAPK)/c-Jun N-terminal kinase (JNK) and diverse caspase cascades, which also contribute to β-cell apoptosis [12].

Saturated palmitate and cis-unsaturated oleate have been widely investigated and a remarkably greater toxicity of the former FA has been repeatedly demonstrated [13,14]. In contrast, very little is known about the cellular effects of dietary trans fatty acids (TFAs), despite their implication in type 2 diabetes mellitus [15]. We have recently compared the effects and metabolism of the two major dietary TFAs, i.e., elaidate (18:1 trans-Δ9) of industrial origin and vaccenate (18:1 trans-Δ11) derived from natural sources, with those of palmitate and oleate in RINm5F rat insulinoma cells. It has been demonstrated that the toxicity of both TFAs were much lower than that of palmitate and similar to that of oleate, which correlated well with the marked accumulation of ceramides and diglycerides (DGs) in palmitate-treated cells and with the mild elevation in ceramide and DG levels upon the administration of cis- or trans-unsaturated FAs [16]. 

Besides the obvious difference between the cell damage caused by saturated and unsaturated FAs, the potentially protective effect of oleate against palmitate toxicity has also drawn growing attention in the past few years. Simultaneously added oleate alleviated palmitate induced toxicity in various cells, including hepatocytes and hepatoma cells [17], human mesenchymal stromal cells and osteoblasts [18], and also decreased non-alcoholic steatohepatitis (NASH) injury in rats with high fat diets [17]. Most importantly, it has been shown that co-administration of oleate can prevent palmitate-induced activation of the unfolded protein response (UPR) in β-cells [19]. In light of our recent observations [16], we find it intriguing if dietary TFAs are also protective like cis-oleate when added to the cells together with palmitate.

The primary aim of the current study was to test the potential interference of elaidate and vaccenate with the palmitate toxicity in RINm5F cells. We also wanted to investigate how oleate, elaidate and vaccenate influence palmitate-induced intracellular accumulation of ceramides and DGs. To this end, we treated the cells with palmitate at toxic concentrations and co-administered one of the unsaturated FAs, then assessed cell viability and the intensity of apoptosis. We also monitored the underlying ER stress and phosphorylation of stress-activated protein kinase (SAPK)/c-Jun N-terminal kinase (JNK). The incorporation and intracellular metabolism of FAs were investigated by determining the amount of the most relevant FA derivatives, as well as of various ceramides and DGs at different times of incubation.

## 2. Results

### 2.1. Effect of Unsaturated Fatty Acids on Palmitate-Induced Cell Death

Investigation of lipotoxicity in RINm5F cells has been established in our laboratory [8,16,20]. Cell viability was assessed by using an MTT assay after 24 h treatments with palmitate or co-treatments with palmitate and an unsaturated FA at 250 or 500 µM individual concentrations. In line with previous findings, palmitate applied at each concentration reduced cell viability to below 40% or below 20% of the control value, respectively (Figure 1). It was also in accordance with earlier studies that cell viability improved significantly upon the simultaneous administration of oleate. The survival rates at least doubled, as almost 80% or 50% of control viability was found after palmitate and oleate co-treatment (Figure 1). Most importantly, the two investigated TFAs, elaidate and vaccenate, both rescued the cells remarkably. Although their protective effect was visibly but not significantly less efficient than that of oleate, they increased viability to above 60% at 250 µM, and to nearly 40% at 500 µM concentrations without any obvious difference between the two of them (Figure 1).

Lipoapoptosis is a major determinant of the palmitate-induced reduction of β-cell viability, and its role has also been demonstrated in our model system. To reveal the contribution of apoptotic intensity to the alterations in cell viability in our experiments, the apoptotic index, i.e., the number of apoptotic cells among 100 total cells was determined at 8 h of FA treatments by using fluorescence microscopy. Palmitate treatment caused a marked five-fold increase in the apoptotic index above that of the control cells, while all the unsaturated FAs exerted significant anti-apoptotic actions, and a mild elevation of apoptotic tendency in the cells treated with any of the FA combinations was found to be statistically insignificant when compared to control cells (Figure 2a).

The observed apoptosis-reducing effect of the unsaturated FAs was further supported by the assessment of caspase-3 activation. The amount of cleaved caspase-3 changed almost parallel to the apoptosis index (Figure 2b). It was hardly detectable by Western blotting in the control cells, and it showed a massive density in the lane of palmitate-treated cell samples. Although the band was still clearly visible, and hence markedly different from the control, oleate co-treatment caused an obvious reduction in caspase-3 activation, and a similar, yet milder, effect was also seen after combinational TFA treatments. Statistical significance was revealed in cases of elaidate and oleate (Figure 2b).

### 2.2. Modulation of Palmitate-Induced Stress

The ER stress and the subsequent UPR are involved in the cellular mechanism of lipotoxicity, as we have also confirmed in RINm5F cells. Moreover, the β-cell protective effect of oleate has also been attributed to its amelioration [19]. Here, we tested the possible contribution of this mechanism of action for all the three unsaturated FAs by examining two early and one late ER stress marker(s). UPR-specific “splicing” of XBP-1 mRNA was analyzed through RT PCR and restriction nuclease cleavage and, in accordance with our previous findings, it revealed a more than five-fold increase in the spliced over unspliced mRNA (sXBP-1/uXBP-1) ratio upon palmitate treatment (Figure 3a). This was significantly counteracted by either of the unsaturated FAs; nevertheless, the effect of oleate appeared to be, again, slightly more pronounced in comparison with the TFAs, and elaidate seemed to be the least effective (Figure 3a).

Phosphorylation of eIF2α is another hallmark of the ER stress, and it is efficiently triggered by palmitate in our model, as was shown previously. This approximately four-fold elevation was almost completely prevented by oleate co-treatment, and it was also diminished by the TFAs, although to a different extent, i.e., vaccenate was almost as potent as oleate, and the effect of elaidate was weaker again (Figure 3b). The adaptive induction of ER chaperones is due to transcriptional regulation by factors produced in the course of the UPR, and hence its development takes longer, and is a widely used marker of sustained ER stress. In line with our earlier report, the 8 h palmitate treatment did not increase the level of the two investigated main ER chaperones, BiP and PDI, in our experiments, and the combinational FA treatments remained ineffective as well (Figure 3b).

Phosphorylation of stress-activated protein kinase/c-Jun N-terminal kinase (SAPK/JNK) plays a central role in lipotoxicity, as the pathways of inflammation, oxidative stress, ER stress and even ceramide accumulation converge at this major stress-activated kinase, which, in turn, contributes to insulin resistance and programmed cell death. Phosphorylated JNK isoforms were detected by immunoblotting and, as expected, a five-fold elevation was seen in palmitate-treated cells compared to bovine serum albumin (BSA)-treated control cells (Figure 3b).

Co-treatment with the cis-unsaturated oleate prevented this effect remarkably, and the phosphorylation level was slightly but not significantly higher than the control background. Interestingly, however, none of the TFAs influenced the palmitate-stimulated JNK phosphorylation significantly; nevertheless, the elaidate caused a 40% decrease compared to palmitate treatment, but it was deemed non-significant by the statistical analysis, while vaccenate did not cause any notable changes (Figure 3b).

### 2.3. Changes in the Fatty Acid Profile of the Cells

Incorporation and metabolism of FAs were monitored at different times during 24 h long treatments. The FA profile was determined using gas chromatography coupled-flame ionization detector (GC-FID) analysis after removal of the medium, saponification of complex cellular lipids and derivatization of the unesterified FAs. All the FAs were incorporated into the cells efficiently. When palmitate was administered alone, its level peaked above 150 μg/mg protein after 16 h of incubation; however, when it was added in combinations, its level seemed to be stabilized around 70%–80 μg/mg (i.e., about 30–40 μg/mg protein above the control level) throughout the whole period of the experiments (Figure 4). The unsaturated FAs given along with palmitate showed a rather similar incorporation, i.e., an increase of 30–40 μg/mg protein beyond the initial background levels, which, in fact, were zero for the two TFAs (Figure 4). Interestingly, a slight decrease in the level of oleate could be observed in the cells treated with palmitate and either of the TFAs, while this phenomenon was missing when palmitate was added alone.

Elongated and desaturated derivatives of palmitate, i.e., stearate (18:0) and cis-palmitoleate (16:1 cis-Δ9), respectively showed obvious elevations when palmitate was added on its own, and these elevations were abolished by simultaneously administered unsaturated FAs. Degradation intermediates of elaidate and vaccenate (16:1 trans-Δ7 and 16:1 trans-Δ9, respectively) were detected in the appropriate cells, while the amount of the corresponding 16:1 cis-Δ7 FA did not increase upon oleate co-treatment (Figure 4).

### 2.4. Diglyceride and Ceramide Levels

The excess acyl-CoA can be channeled towards triglyceride (TG) synthesis; however, the uneven supply of saturated and unsaturated FAs might cause a hindrance in this route at DG intermediates. Therefore, the most important DG species containing 16 and/or 18 carbon long saturated and/or monounsaturated FAs were assessed by LC-MS/MS. The most striking elevations were observed after 8 h long palmitate treatment, when the levels of 1,2 dipalmitoyl-glycerol (16:0/16:0), 1-palmitoyl-2-stearoyl-glycerol (16:0/18:0), 1,2-distearoyl-glycerol (18:0/18:0), 1-palmitoyl-2-palmitoleoyl-glycerol (16:0/16:1) and 1-palmitoyl-2-oleoyl-glycerol (16:0/18:1) were increased 45-fold, 18-fold, eight-fold, five-fold and three-fold, respectively while the levels of 1-stearoyl-2-oleoyl-glycerol (18:0/18:1) and 1,2-dioleoyl-glycerol (18:1/18:1) remained practically unaffected (Figure 5 and Appendix A). Altogether, palmitate treatment caused a 17-fold increase in the overall DG levels compared the control cells (Figure 5 and Appendix A).

Despite an inevitable increase in the overall DG content upon palmitate plus oleate co-treatment, the elevation was about two-fold, which remained far below that caused by palmitate alone. The time-course and the pattern of the effect were also different as the overall DG levels were kept stable from 4 to 24 h of incubation, and the contribution of DG species containing 18 carbon long unsaturated chains were similar to those containing only palmitoyl, palmitoleoyl and/or stearoyl moieties (Figure 5 and Appendix A). Co-treatments with one of the two TFAs induced yet another course of DG accumulation. The overall DG levels kept increasing above the control and even above the oleate co-treated samples for 16 h, and they only seemed to settle at the end of the incubations when they were approximately twice as high as in the oleate co-treated samples and actually almost reached the already decreasing overall DG concentration of the palmitate-treated cells. Nevertheless, the compositions were also characteristically different, because the amount of DG species containing 18 carbon long unsaturated chains grew uniquely high in the presence of elaidate or vaccenate, and these DG molecules became a dominant fraction of the total (Figure 5 and Appendix A). A comparison between the two TFAs shows that elaidate co-treatment yielded higher DG levels in the longer incubations, and this was due mostly to a more pronounced increase in the amount of DG species containing 18 carbon long unsaturated chains (Figure 5 and Appendix A).

Ceramides, the biosynthetic lipid intermediates of proven contribution to β-cell lipotoxicity, were also assessed at different times of incubation by LC-MS/MS measurement. Palmitate treatment caused a marked and sustained accumulation of ceramides containing saturated FA chains, either palmitate or stearate. The overall ceramide content in these cells was about five times the control value (5.70 vs. 1.22 μg/mg protein) at 24 h, while the amount of oleoyl–sphingosine (18:1) remained negligible, i.e., it only increased from 9.5 to 16.6 ng/mg protein (Figure 6 and Appendix A). The presence of oleate efficiently attenuated this effect by reducing the palmitate-induced accumulation of both palmitoyl– and stearoyl–sphingosines (16:0 and 18:0); nevertheless, despite oleate abundance, oleoyl–sphingosine (18:1) was still hardly detectable, though its level increased slightly to 21.9 ng/mg protein at 24 h (Figure 6 and Appendix A). The two TFAs acted similarly but not equivalently to oleate as, on the one hand, they were nearly as effective in diminishing palmitoyl– and stearoyl–sphingosines (16:0 and 18:0) as oleate was and, on the other hand, they caused a remarkable increase in elaidyl– or vaccenyl–sphingosine (18:1) concentrations, which reached to as high as 487 ng/mg protein and 448 ng/mg protein at 24 h (Figure 6 and Appendix A).

## 3. Discussion

The FA composition of dietary lipids has been long known to influence health. The advantage of monounsaturated and polyunsaturated FAs over the saturated ones is widely accepted, and extensive research has aimed to seek experimental evidence as well as to reveal the molecular mechanism of the phenomenon. Not only were the unsaturated FAs found to be less toxic in cultured or isolated cells, but they also alleviated the damage caused by saturated FAs in certain cell types, including β-cells. The spread of the industrial processing of plant oils shed light on the biological importance of configurational isomerism at the double bonds in unsaturated FAs. Some natural cis double bonds assume the trans arrangement through spontaneous isomerization during industrial hydrogenation. These TFAs have been declared to be the worst types of FAs, and are held at least partly responsible for the health risks associated to the consumption of commercial cookies and pastries or fast-food French fries. In this case, however, the need for scientific thoroughness seems to be considerably weaker, and TFAs have been banned in certain countries on the basis of partly contradictory in vivo observations and without any convincing evidence of extreme toxicity at cellular or molecular levels. It is also peculiar that TFAs ingested with ruminant meat and dairy products, which are produced by bacterial isomerization, are exonerated from the same accusations despite their similar structure.

We aimed to compare the effects of prototypical saturated, cis- and trans-unsaturated FAs in cellular systems with special attention paid to the induction of ER stress and apoptosis. Using an established model of lipotoxicity in an insulinoma cell line [8], we demonstrated that the industrial elaidate and natural vaccenate were equally less toxic than the saturated palmitate in our short-term in vitro experiments. The only noteworthy difference between the above-mentioned TFAs and cis-oleate was in their metabolism, i.e., although the accumulation of ceramides and DGs upon the addition of any unsaturated FA remained negligible compared to that induced by palmitate, the ceramide and DG accumulation caused by either of the two TFAs was remarkably greater than that caused by oleate [16]. The difference was clearly due to a build-up of ceramide and DG molecules containing monounsaturated FAs, and hence we speculated that either TFAs might be more favorable substrates for ceramide and DG synthesis or the ceramides and DGs containing TFA chains might be less favorable substrates for further conversions. Since TG deposition could not be seen in the electron micrographs, nor could it be detected by light microscopy after neutral fat staining, this question remains to be elucidated. Regardless, the accumulation of ceramides and DGs has a large biological significance, especially in the long-term β-cell functionality and viability, and thus in the development of diabetes mellitus.

In the present study, we continued our research by comparing the ability of elaidate and vaccenate to attenuate palmitate toxicity with that of oleate in the same cellular model. BSA-conjugated palmitate was added to the cells at toxic concentrations (250 and 500 μM), which have been seen to severely reduce cell viability (by about 60% and 85%, respectively) at 24 h, while the obvious signs of apoptosis induction and stress, including ER stress, were investigated at 500 μM palmitate concentration and after 8 h incubation. Each of the three unsaturated FAs was administered simultaneously with palmitate and at the same concentration. In spite of a double FA dose, the cellular damage caused by a palmitate and oleate combination remained far below that induced by palmitate alone, which was in accordance with previous findings. Cells were efficiently rescued by oleate co-treatment from lipotoxic death, in particular, from lipoapoptosis, and from the development of stress, as assessed by detection of JNK phosphorylation, as well as from the activation of the UPR, i.e., XBP-1 mRNA cleavage and eIF2α phosphorylation. Ceramides and DGs are biosynthetic lipid intermediates that are implicated in the deleterious consequences of FA overload, specifically in ER stress and apoptosis, and particularly in β-cells. In line with our previous findings, oleate was scarcely incorporated in ceramides; moreover, the presence of oleate largely reduced the palmitate-induced elevation in the level of those ceramides that contain saturated (palmitoyl or stearoyl) FA chains, and thus oleate was able to minimize the overall palmitate-induced ceramide accumulation. Although oleate incorporation into DGs was much more pronounced, a similar effect was seen with regards to this group of lipid intermediates, as the amount of DG species containing saturated chains, and also the overall amount of DGs, increased remarkably less when palmitate treatment was combined with oleate addition. Altogether, the expected protective effect of oleate against palmitate toxicity was evident in our experiments, and it correlated very well with the mitigation of ceramide and DG accumulation in the cells.

The two investigated TFAs were also co-administered with palmitate, and they exerted a rather similar protection against the deleterious effects of the saturated FA. Interestingly, the level of JNK phosphorylation, indicative of a general stress intensity in the cells, was not diminished by the TFAs as effectively as it was by oleate, and the attenuation of ER stress by elaidate was remarkably weaker compared to the other two unsaturated FAs; these differences were not reflected by cell viability and apoptosis assessments. The marked reduction in cell death corresponded with the obvious alleviation of ceramide build-up in the presence of unsaturated, either cis or trans, FAs. Although a detectable amount of unsaturated ceramides appeared in the TFA co-treated cells, the presence of TFAs prevented the palmitate-induced accumulation of ceramides containing saturated FA chains, and minimized the overall palmitate-induced ceramide accumulation like oleate did. A similar phenomenon can be observed with regards to DGs in the first 8 h of the incubations; however, the total DG levels in palmitate-treated and palmitate and TFA co-treated cells converge in longer incubations, partly due to a marked growth in the species containing TFA chains. These data further support the view that ceramides play a central role in the development of β-cell lipotoxicity and lipoapoptosis. They also indicate that, at least as long as the β-cells and diabetes are concerned, TFAs are not the worst and most harmful types of FAs, and they rather fall between the most deleterious saturated and the most protective cis-unsaturated ones.

The outstanding toxicity of saturated FAs is often explained by the reluctance of the cells to process fully saturated DGs, and by the consequent ineffectiveness of channeling the saturated acyl-CoA surplus towards TG synthesis [21]. The theory that unsaturated FAs attenuate palmitate toxicity by facilitating TG synthesis is supported by experimental data obtained in certain cell types, such as CHO cells and primary mouse embryonic fibroblasts [22], mouse myoblasts [23], human mesenchymal stromal cells and osteoblasts [18]. Nevertheless, a palmitate treatment was used in HepG2 hepatocarcinoma cells [17] and controversial data have been published as to the deleterious or protective role of TG deposition in β-cell lipotoxicity [24,25]. A comparison of the FA profiles suggest that this proposed mechanism might not play a central role in the protective effect of the three unsaturated FAs observed in our experiments. The facilitated diversion of palmitoyl-CoA towards TG synthesis would be expected to enhance palmitate incorporation into the cells. However, co-treatments of our insulinoma cells with unsaturated FAs decreased the amount of palmitate with about as much as the increment in the amount of these unsaturated FAs. Together with the above-mentioned lack of detectable lipid droplet (TG) deposition, this finding suggests that the reduction in ceramide and DG accumulation can be attributed, at least partly, to a lower palmitate uptake in the co-treated cells. The possible interference between the cellular uptake of different FAs and/or the saturation of FA uptake in our experimental conditions deserves further investigation.

In summary, here we demonstrate the prevention of palmitate-induced excessive ceramide and DG accumulation in RINm5F cells by elaidate and vaccenate. The observed reduction in the levels of these potentially harmful biosynthetic lipid intermediates correlated well with the protective effect of the two investigated dietary TFAs.

## 4. Materials and Methods 

### 4.1. Materials Used

The culture medium and supplements were purchased from Thermo Fisher Scientific (Waltham, MA, USA). Palmitate, oleate, elaidate, vaccenate, FA free bovine serum albumin (BSA), trans-vaccenic acid methyl ester, methyl oleate methyl palmitate, methyl palmitoleate methyl stearate, 1,2-dipalmitoyl-rac-glycerol (>99%), 1-palmitoyl-2-oleoyl-sn-glycerol (>99%), 1,2-dioleoyl-sn-glycerol (>97%) and 1-octadecanoyl-2-hexandecanoyl-sn-glycerol (>99%) were from Sigma Aldrich (St. Louis, MO, USA), n-hexane was purchased from Romil (Waterbeach, UK). C16:0, C17:0, C18:0, C18:1(9Z) (>99%) ceramides were purchased from Avanti Polar Lipids Inc. (Alabaster, AL, USA).Methanol (gradient grade) and acetonitrile (gradient grade) were purchased from Merck KGaA. (Darmstadt, Germany). All other chemicals used in this study were of analytical grade. All experiments and measurements were carried out by using Milipore (Darmstadt, Germany) ultrapure water.

### 4.2. Cell Culture and Treatment

RINm5F rat insulinoma cells were purchased from ATCC and cultured in RPMI 1640 medium, containing 2 mM L-glutamine, 1.5 g/l sodium bicarbonate, 4.5 g/l glucose, 10 mM HEPES and 1 mM sodium pyruvate and supplemented with 10% fetal bovine serum and 1% antibiotics (Thermo Fisher Scientific; Waltham, MA, USA), at 37 °C in humidified atmosphere containing 5% CO_2_.

Palmitate, elaidate, oleate and vaccenate were diluted in isopropanol (Molar Chemicals, Halásztelek, Hungary) to a concentration of 50 mM, conjugated with 4.16 mM FA free BSA in 1:4 ratio, at 37 °C for 1 h. The working solution for FA treatments was always prepared freshly in fetal bovine serum (FBS)-free and antibiotic-free medium at 0.25 or 0.5 mM final concentration. The culture medium had been replaced by FBS-free and antibiotic-free medium for 1 h before the cells were treated with FA for 4–24 h at 70%–80% confluence in 6-well plates (for Western blot, RT-PCR and analysis of ceramides, DGs and FA profile) or in 96-well plates (for cell viability assay and detection of apoptosis and necrosis).

### 4.3. Cell Viability and Apoptosis

Cell viability was assessed by using the Colorimetric (MTT) Kit for Cell Survival and Proliferation (Merck Kft.; Darmstadt, Germany) according to the manufacturer’s instructions. MTT-derived formazan was measured at 530 nm test and 630 nm reference wavelengths in a multiscan spectrophotometer (Thermo Fisher Scientific; Waltham, MA, USA). Cell viability was expressed as the percentage of viable cells in the total cell population.

Apoptotic and necrotic cells were detected by using Annexin-V-FLUOS Staining Kit (Roche; Basel, Switzerland) and fluorescence microscopy according to the manufacturer’s instructions. Cells with green fluorescence (Annexin V labeling) were considered as apoptotic while those with red or both green and red fluorescence (propidium iodide DNA staining) were considered as necrotic. A minimum of 1000 cells was counted in each experimental condition. Apoptosis index was calculated as (number of apoptotic cells) / (number of all cells counted) × 100.

### 4.4. Western Blot Analysis

Cells were washed twice with PBS and harvested in 100 µL lysis buffer by scraping. The lysis buffer contained 0.1% SDS, 5 mM EDTA, 150 mM NaCl, 50 mM Tris, 1% Tween 20, 1 mM Na_3_VO_4_, 1 mM PMSF, 10 mM benzamidine, 20 mM NaF, 1 mM pNPP and protease inhibitor cocktail. The lysates were centrifuged in a benchtop centrifuge (10 min, 10,000 rpm, 4 °C). Protein concentration of the supernatant was measured using Pierce BCA Protein Kit Assay (Thermo Fisher Scientific; Waltham, MA, USA), and the samples were stored at -20 °C until use.

Samples (20 μg protein) were electrophoresed in 10%–12%–15% SDS polyacrylamide gels and transferred to PVDF membranes (Millipore; Darmstadt, Germany). Primary and secondary antibodies were applied overnight at 4 °C and for 1 h at room temperature, respectively. Equal protein loading was validated by detection of glyceraldehyde 3-phosphate dehydrogenase (GAPDH), with a mouse monoclonal anti-GAPDH (Santa Cruz; Dallas, TX, USA, sc-32233) antibody, at 1:20,000 dilution, as a constitutively expressed reference protein. Primary antibodies: rabbit anti-Cleaved Caspase-3 (#9661), rabbit anti-phospho-SAPK/JNK (THR183/Tyr185) (#9251S), rabbit anti-SAPK/JNK (#9252S), rabbit anti-phospho-eIF2α (#9721), rabbit anti-eIF2α (#9722) from Cell Signaling (Danvers, MA, USA), goat anti-GRP78 (sc-1050) and rabbit anti-PDI (sc-20132) from Santa Cruz (Dallas, TX, USA). Secondary antibodies: horseradish peroxidase (HRP)-conjugated goat anti-rabbit IgG-HRP (#7074), HRP-conjugated horse anti-mouse IgG-HRP (#7076) from Cell Signaling (Danvers, MA, USA) and donkey anti-goat IgG-HRP (sc-2020) from Santa Cruz (Dallas, TX, USA). HRP was detected with chemiluminescence using SuperSignal West Pico Chemiluminescent Substrate (Thermo Fisher Scientific; Waltham, MA, USA).

### 4.5. RT-PCR and Endonuclease Cleavage

Total RNA was purified from the cells by using RNeasy Plus Mini Kit (Quagen; Germantown, MD, USA) following the manufacturer’s instruction. cDNA was produced by reverse transcription of 0.5 μg DNA-free RNA samples using SuperScript III First-Strand Synthesis System for RT-PCR Kit (Invitrogen; Carlsbad, CA, USA). Spliced and unspliced XBP-1 sequences (421 or 447 bp, respectively) were amplified by PCR using iProof High-Fidelity DNA Polymerase Kit (BioRad; Hercules, CA, USA) and 5’ - GCT TGT GAT TGA GAA CCA GG - 3’ SY121041268-007 XBP-1 sense (rat) and 5’ - AGG CTT GGT GTA TAC ATG G - 3’ ST00450236-001 XBP-1 antisense (mouse, rat) primers (Sigma Aldrich (St. Louis, MO, USA) at thermocycle conditions of 98 °C 3 min followed by 30 cycles of 98 °C 10 s, 57 °C 30 s and 72 °C 15 s and finished by 72 °C 10 min final extension. PCR products were purified by PEG precipitation. DNA concentration in the PCR products was measured with NanoDrop 1000 Spectrophotometer (Thermo Fisher Scientific; Waltham, MA, USA). For a better visibility, PstI restriction endonuclease cleavage of 200 ng purified PCR product was carried out by using FastDigest PstI (Thermo Fisher Scientific; Waltham, MA, USA) for 30 min at 37 °C. The unspliced XBP-1 is cut into two fragments (153 and 294 bp) by PstI while the spliced variant remains uncut [8]. A 234 bp fragment of the GAPDH was also amplified as a reference cDNA using a GAPDH sense primer 5’ – AGA CAG CCG CAT CTT CTT GT - 3′ and a GAPDH antisense primer 5’ - CTT GCC GTG GGT AGA GTC AT - 3’. The PCR thermocycle conditions were 98 °C 1 min, then 28 cycles of 98 °C 10 s, 65 °C 30 s and 72 °C 30 s, completed by a final extension of 72 °C 10 min. Equal amounts of DNA samples (i.e., digested XBP-1 or GAPDH) were separated by electrophoresis in 2% agarose gel and visualized by EtBr staining.

### 4.6. Analysis of Lipid Contents

For FA, ceramide and DG analysis, cells were washed once with PBS, then harvested in 100 µL PBS by scraping. The samples were then sedimented in a benchtop centrifuge (5 min, 1,500 rpm, 24 °C), and the supernatants were discarded. The cells were suspended in PBS, and the protein concentration of the cell suspension was measured as mentioned at Western blot analysis. A total of 50 µL of each suspension was transferred to a clear crimp vial for GC-FID measurement of FAs, and 50 µL was transferred to a micro-centrifuge tube for the HPLC-MS/MS analysis of DGs and ceramides.

#### 4.6.1. GC-FID Analysis of Fatty Acid Profiles

150 µL of methanol containing 2 W/V% NaOH was added to the 50 µL cell suspension in the crimp vials, the samples were incubated at 90 °C for 30 min, and then cooled to room temperature. A total of 400 µL of methanol containing 13%–15% of boron trifluoride was added to the samples, and the vials were incubated at 90 °C for 30 min. After cooling to room temperature, 200 µL of saturated NaCl solution and 300 µL of n-hexane were added. FA methyl esters were extracted to the upper phase containing n-hexane, and this phase was transferred to a vial for GC analysis.

Samples of 1 µL volume were separated in a Zebron ZB-88 capillary column (60 m x 0.25 mm i.d., 0.20 µm film thickness) (Phenomenex, Torrance, CA, USA) by using a Shimadzu GC-2014 gas chromatograph equipped with a Shimadzu AOC-20s autosampler and a flame ionization detector (FID) (Shimadzu, Kyoto, Japan). The carrier gas was hydrogen at 35 cm/sec velocity. The injector and detector temperature was 250 °C, and the oven temperature was ramped from 100 °C to 210 °C at a rate of 4 °C/min.

#### 4.6.2. HPLC-MS/MS Analysis of Diglycerides and Ceramides

The cells were pelleted by centrifugation (5 min, 1,500 rpm, 24 °C) and resuspended in methanol containing ceramide 17:0 internal standard (50 ng/ml). The samples were agitated with an ultrasonic sonotrode and centrifuged (10 min, 13,400 rpm, 24 °C). The supernatants were transferred to vials for HPLC-MS/MS analysis.

Samples (10 µL) were injected in the HPLC composed of a Perkin Elmer series 200 high-pressure gradient pump, autosampler, online degasser and a thermostat (Perkin Elmer, Milan, Italy). A Kinetex^®^ 5 µm, C8 100 Å, LC (100 x 3 mm) column (Phenomenex, Torrance, CA, USA) was used with a gradient elution of methanol (mobile phase A) and 10 mM ammonium-acetate (mobile phase B): 0 min at 90% A; 1 min at 90% A; 9 min at 95% A; 10.5 min at 98% A; 11.5 min at 98% A; 12 min at 90% A; 14 min at 90% A. Ceramide and diglyceride species were detected using a triple quadrupole mass spectrometer (Applied Biosystems MDS SCIEX 4000 Q TRAP) (Sciex, Framingham, MA, USA). The instrument was used in positive multiple reaction monitoring mode. The ion spray temperature was set to 400 °C and the voltage to 5500 V.

### 4.7. Statistics

The results of Western blot analyses and DNA gel electrophoresis were carried out by densitometry using ImageQuant 5.2 software and are shown as relative band densities normalized to GAPDH as a reference protein. Data are presented in the diagrams as mean values ± S.D. and were compared by ANOVA with Tukey’s multiple comparisons post-hoc test using GraphPad (San Diego, CA, USA) Prism 6 software. Differences in *p* values below 0.05 were considered to be statistically significant.

## 5. Conclusions

Our results show the remarkable protective effect of TFAs against palmitate-induced short-term toxicity, particularly ER stress and apoptosis in insulinoma cells on the one hand, yet a marked and important difference between the utilization of TFAs and oleate in ceramide and DG synthesis, with probable long-term health impacts on the other hand. The molecular background behind the preferred incorporation of trans- vs. cis-unsaturated FAs into ceramides as well as the potential inhibition of palmitate uptake by simultaneously administered unsaturated FAs remain to be elucidated.

## Figures and Tables

**Figure 1 ijms-21-02626-f001:**
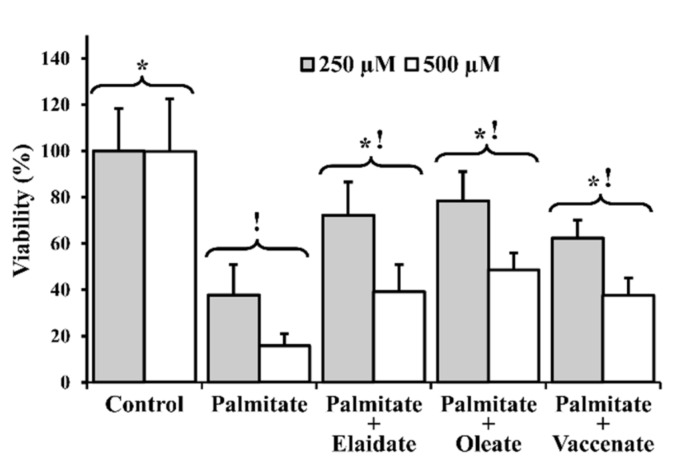
Cell viability. RINm5F insulinoma cells were treated with bovine serum albumin (BSA) (control cells) or BSA-conjugated palmitate alone, or with palmitate in combination with one of the unsaturated fatty acids, elaidate, oleate or vaccenate at two concentrations (250 or 500 µM) as indicated at 70%–80% confluence and incubated for 24 h. Cell viability was measured by Colorimetric (MTT) Kit for Cell Survival and Proliferation (Millipore) and expressed as a percentage of the control. Data are shown as mean values ± S.D; *n* = 6; statistically significant differences: ^!^
*p* < 0.05, vs. BSA-treated control; * *p* < 0.05, vs. palmitate-treated cells.

**Figure 2 ijms-21-02626-f002:**
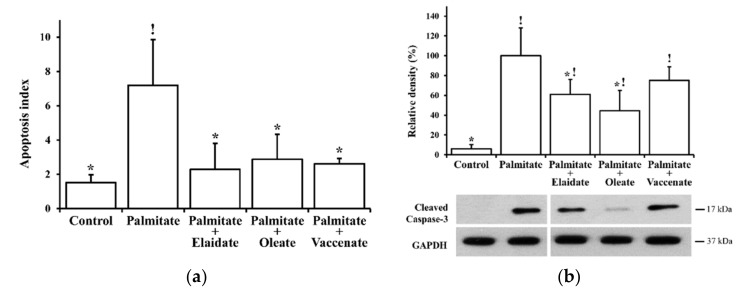
Lipoapoptosis. Cells were treated with BSA (control cells) or BSA-conjugated palmitate alone, or with palmitate in combination with one of the unsaturated fatty acids, elaidate, oleate or vaccenate at 500 µM individual concentrations for 8 h at 70%–80% confluence. (**a**) Apoptotic cells/bodies were detected and counted by annexin and propidium iodide staining and fluorescence microscopy. Apoptosis index was calculated as the relative number of apoptotic cells and expressed as percentage of the total cell number. (**b**) Cleaved caspase-3 was detected by Western blot in cell lysates. The image shows typical results of three independent experiments with two parallels. The results were quantified by densitometry, normalized to glyceraldehyde 3-phosphate dehydrogenase (GAPDH) as a constitutive reference protein and are shown as relative band densities in the percentage of palmitate values. Data in all diagrams are shown as mean values ± S.D; *n* = 6; statistically significant differences: ^!^
*p* < 0.05, vs. BSA-treated control; * *p* < 0.05, vs. palmitate-treated cells.

**Figure 3 ijms-21-02626-f003:**
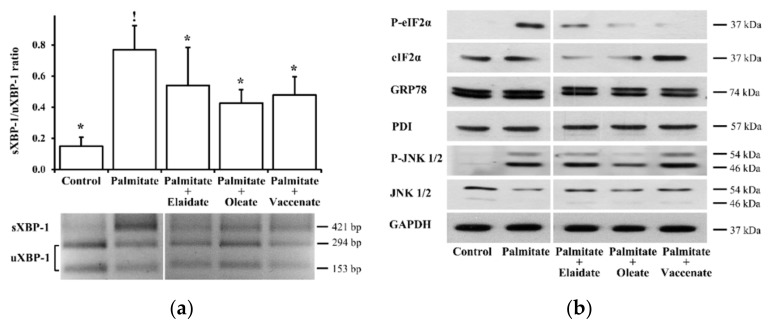
Stress markers. Cells were treated with BSA (control cells) or BSA-conjugated palmitate alone, or with palmitate in combination with one of the unsaturated fatty acids, elaidate, oleate or vaccenate at 500 µM individual concentrations for 8 h at 70%–80% confluence. (**a**) Total RNA was prepared and a 447 or 421 bp long sequence was amplified by RT-PCR from “spliced” sXBP-1 or “unspliced” uXBP-1 mRNA versions, respectively. A better visualization was aimed for by PstI restriction endonuclease digestion, which leaves sXBP-1 cDNA uncut (421), while cleaves uXBP-1 cDNA to two fragments (153 and 294 bp). The cDNAs were separated by 2% agarose gel electrophoresis, the band densities were quantified by densitometry, and sXBP-1/uXBP-1 density ratios were calculated. A typical gel image obtained in one of three independent experiments is presented. (**b**) Phosphorylated eIF2α (P-eIF2α), 78 kDa glucose-regulated protein (GRP78/BiP), protein disulfide isomerase (PDI), phosphorylated SAPK/JNK isoforms (P-JNK 1/2) protein levels were detected by Western blot in the cell lysates. The images show typical results of three independent experiments with two parallels. The results were quantified by densitometry, normalized to GAPDH levels and are presented as relative band densities in the percentage of palmitate-treated samples. Data are shown as mean values ± S.D.; *n* = 6; ^!^
*p* < 0.05, vs. BSA-treated control; * *p* < 0.05, vs. palmitate-treated cells.

**Figure 4 ijms-21-02626-f004:**
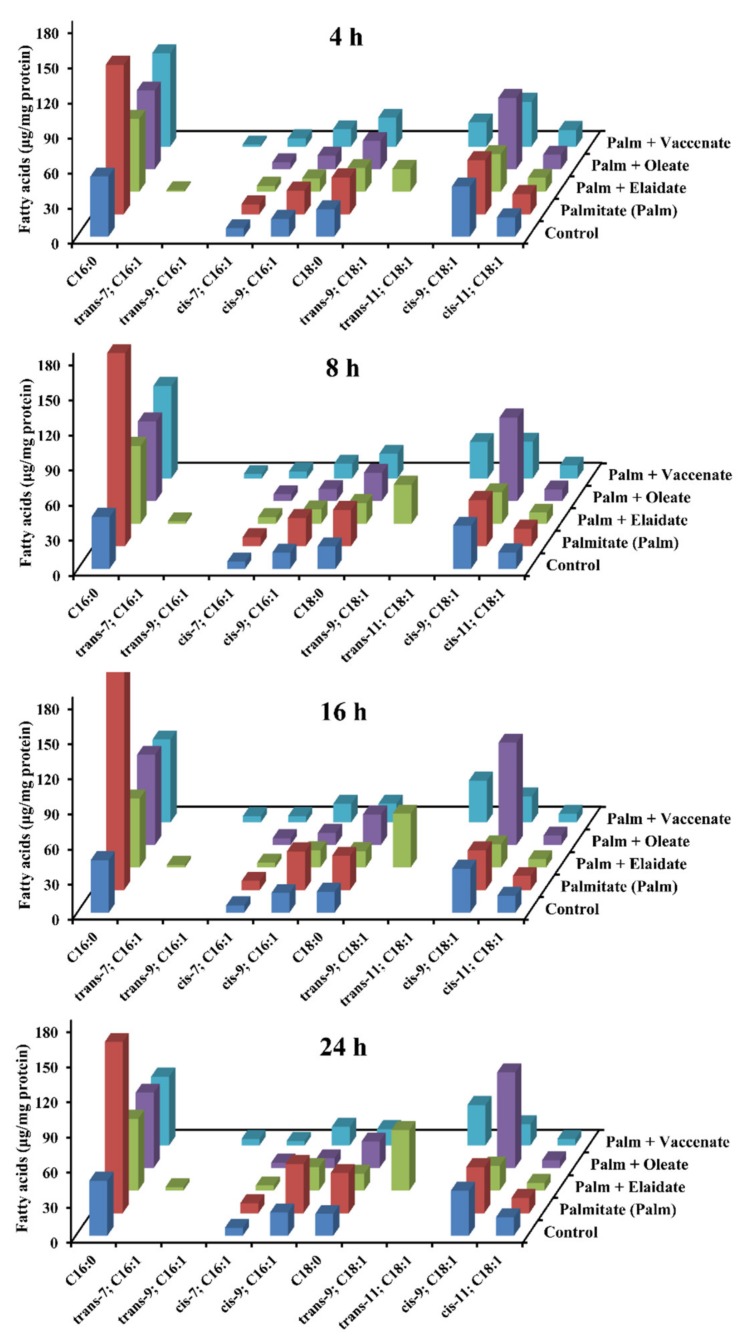
Fatty acid profile. Cells were treated with BSA (control cells) or BSA-conjugated palmitate alone, or with palmitate and one of the unsaturated fatty acids, elaidate, oleate or vaccenate at 250 µM individual concentration for 8 h at 70%–80% confluence. Washed cell samples were withdrawn at 4, 8, 16 and 24 h times of incubation, and the amounts of ten different saturated and monounsaturated fatty acids were measured by GC-FID after saponification and methylation. Data were normalized to the protein content of the samples, and are shown as mean values of three independent experiments with two parallels.

**Figure 5 ijms-21-02626-f005:**
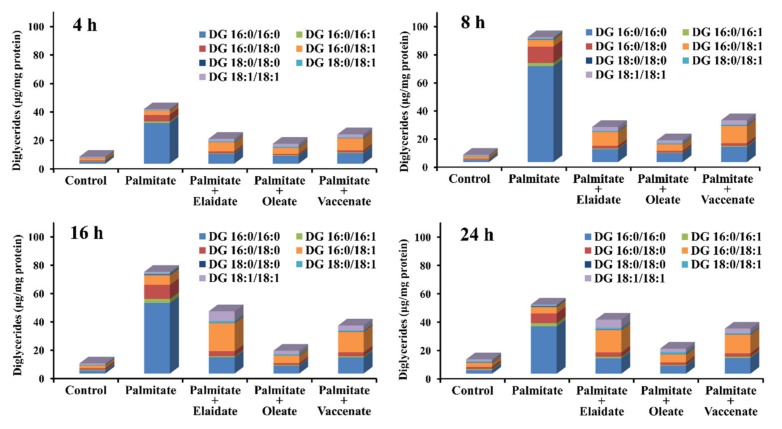
Diglyceride accumulation. Cells were treated with BSA (control cells) or BSA-conjugated palmitate alone, or with palmitate and one of the unsaturated fatty acids, elaidate, oleate or vaccenate at 250 µM individual concentration for 8 h at 70%–80% confluence. The amount of seven major diglyceride species was measured by LC-MS/MS in the washed cell samples prepared after 4, 8, 16 or 24 h treatments. The detected diglycerides contained the indicated combinations of palmitate (16:0), stearate (18:0), and a monounsaturated fatty acid of 16 (16:1) or 18 carbons (18:1). Data were normalized to the protein content of the samples, and are shown as mean values of three independent experiments with two parallels.

**Figure 6 ijms-21-02626-f006:**
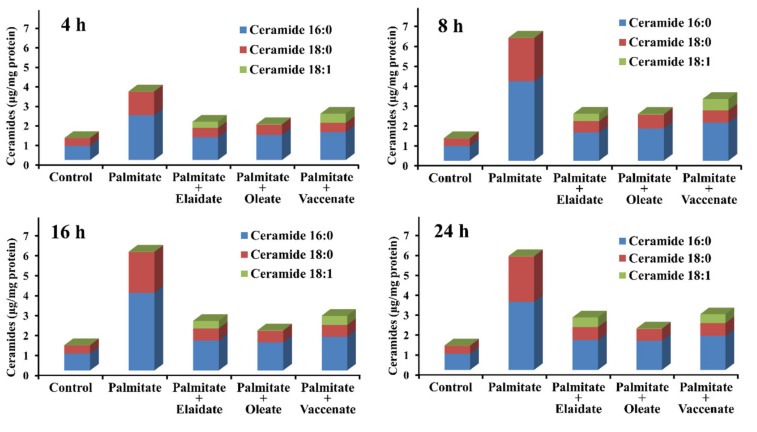
Changes in ceramide levels. Cells were treated with BSA (control cells) or BSA-conjugated palmitate alone, or with palmitate and one of the unsaturated fatty acids, elaidate, oleate or vaccenate at 250 µM individual concentration for 8 h at 70%–80% confluence. The amount of three different ceramide species was measured by LC-MS/MS in the washed cell samples prepared after 4, 8, 16 or 24 h treatments. The detected ceramides contained either a palmitate (16:0) or a stearate (18:0) or a monounsaturated fatty acid of 18 carbons (18:1). Data were normalized to the protein content of the samples, and are shown as mean values of three independent experiments with two parallels.

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
