# Peer review of "Effect of cis- and trans-Monounsaturated Fatty Acids on Palmitate Toxicity and on Palmitate-induced Accumulation of Ceramides and Diglycerides"

_ijms, 2020, doi:10.3390/ijms21072626_

Round 1

Reviewer 1 Report

The aim of the current study is to test the Effect of cis- and trans-monounsaturated fatty acids on palmitate toxicity and on palmitate-induced accumulation of ceramides and diglycerides. The study is nicely designed and totally descriptive. To complete the description, it would be desirable to interpret why TFAs allievate palmitate-induced ER stress and apoptosis and induce accumulation of ceramides and diglycerides in the same time (24 hour)? Many reports indicated that ceramides and diglycerides induce apoptosis and ER stress. The results are conflicting.

Author Response

Thank you very much for reviewing our manuscript, and special thanks for the supportive evaluation.

Saturated palmitate is the most damaging fatty acid in a wide variety of cell lines, including the insulinoma cells used in our study. It caused a massive increase in apoptosis and triggered several markers of stress, particularly those of ER stress within 8 h, and it also dramatically reduced the number of viable cells (to 20% of control) in 24 h. These deleterious effects well correlated with the accumulation of ceramides and diglycerides in the cells, which was in accordance with earlier reports indicating that these lipid species induce apoptosis and ER stress.

Effects of co-administered TFAs on palmitate-induced ER stress and apoptosis were investigated at the first time in our study, and our results further strengthened the proposed role of ceramides and diglycerides in palmitate toxicity. Co-administration of TFAs markedly alleviated palmitate-induced ER stress and apoptosis, and it also reduced palmitate-induced accumulation of ceramides and diglycerides at all times of measurement. It is true that reduction in diglyceride accumulation gradually weakened during co-treatments, it was still obvious at 24 h. Moreover, the cis unsaturated oleate, which was even more effective than the TFAs in protecting the cells from palmitate-induced damages, was also more effective in reducing palmitate-induced accumulation of ceramides and diglycerides. So, TFAs did not cause ceramide and diglyceride build-up, they just reduced the accumulation of these lipid intermediates less efficiently as oleate did.

We hope that our answer resolves the conflict described in the comment, and you find our manuscript acceptable for publication.

Reviewer 2 Report

This work evaluates the protective effect of cis- and trans-monousaturated fatty acids on palmitate toxicity. The investigation is new and original but there are major concerns in regards to the interpretation of the results:

  1. It is likely that the inhibitory effect of monounsaturated FAs is mediated by an artifact linked to the direct binding and aggregation to palmitate
  2. Comparison in regards to protein levels is performed between different blots an this is not appropriate. Samples must be analyzed in the same WB
  3. There are no controls with elaidate, oleate or vuccenate only treatments
  4. What is the apoptotic index? Necrotic, apoptotic, both? late necrotic? The data is poorly analyzed

Author Response

Thank you for taking the time to review our manuscript.

Let us respond to the concerns raised in the comments.

  • It is likely that the inhibitory effect of monounsaturated FAs is mediated by an artifact linked to the direct binding and aggregation to palmitate

All fatty acids were administered in conjugation wits serum albumin (BSA), which is a standard protocol for cellular fatty acid treatments. This preparation of fatty acids, i.e. their pre-incubation with BSA in 1:4 ratio, at 37 °C for 1 hour, yields a stable aqueous solution, in which the vast majority of fatty acids is adhered to albumin molecules. This is how fatty acids can be taken up the most effectively and in the most physiological way by the cultured cells. When two types of fatty acids were applied simultaneously, their BSA-conjugated solutions were prepared added to the cells separately, so the fatty acid – BSA ratio remained unchanged during the experiments. Therefore, we do not see any reason to assume that co-administration might have caused fatty acid aggregation. Nevertheless, we totally agree with the reviewer that the possibility of any artifacts that might interfere with fatty acid accessibility needs to be ruled out when undertaking cell treatments. This is the main reason why we assessed the incorporation of each fatty acid by detecting the cellular fatty acid profiles. These measurements revealed that all the four investigated fatty acids were taken up by the cells effectively, and the overall fatty acid uptake was largely elevated both in single (palmitate only) and in double (palmitate plus a monounsaturated FA) treatments. These findings clearly show that simultaneous addition of two types of FAs did not cause aggregation in the medium.

  • Comparison in regards to protein levels is performed between different blots an this is not appropriate. Samples must be analyzed in the same WB

It is absolutely right that protein levels must be compared only within the same WB due to inevitable differences between the signal developments. We complied to this principle as all the WB images analyzed in this study contained the control (untreated) samples as well as the samples from cells treated with palmitate only, palmitate plus elaidate, palmitate plus oleate and palmitate plus vaccenate together. These complete WB images were uploaded together with the original submission of the manuscript according to the Journal’s requests. However, the WBs also contained some additional samples between the palmitate only and combinational treatments, which were cut out from the pictures presented in figures 2 and 3, and hence the dividing cut lines. In other words, the pictures shown in our manuscript and analyzed in our studies are not merged WB results but incomplete WB images.

  • There are no controls with elaidate, oleate or vuccenate only treatments

We agree that the mentioned single treatments are indeed relevant for the present study as controls. However, when we actually carried out these single treatments we realized that they are not mere controls for the combinational treatments but constitute an individual study on their own. Since the cellular effects and metabolism of trans fatty acids have not been investigated yet, the comparison of palmitate, oleate, elaidate and vaccenate single treatments revealed novel and interesting data, which were published in a recent paper (reference 15 in the present manuscript). Presenting these controls, therefore, would be a duplicate publication of research findings and could be considered as violation of the code of ethics. This is why we provided detailed description of those results in the Introduction (lines 47-53) and Discussion (lines 263-278) sections, and we hope that this ethical reciting of our previous article can substitute for the assailable incorporation of already published data.

  • What is the apoptotic index? Necrotic, apoptotic, both? late necrotic? The data is poorly analyzed

Description of apoptosis index is included in the legend of figure 2: “Apoptosis index was calculated as the relative number of apoptotic cells and expressed as percentage of the total cell number.” and in the Materials and Methods section: “Apoptosis index was calculated as (number of apoptotic cells) / (number of all cells counted) × 100”. It is also defined in the latter that “Apoptotic and necrotic cells were detected by using Annexin-V-FLUOS Staining Kit (Roche) and fluorescence microscopy according to the manufacturer’s instructions. Cells with green fluorescence (Annexin V labeling) were considered as apoptotic while those with red or both green and red fluorescence (propidium iodide DNA staining) were considered as necrotic. A minimum of 1000 cells was counted in each experimental condition.”

We hope that you find our answers acceptable and you give your support to the publication of our manuscript.

Round 2

Reviewer 2 Report

The investigators have addressed my concerns in regards to the rigor of the research. However, some major concerns still exist:

1) The conclusions are still not fully supported by the data. The authors need to demonstrate that indeed caspase-dependent, ER-dependent and JNK-dependent cell death is triggered by Palmitate. This requires straightforward experiments using pharmacological regulators of these cascades.

2) Figure 5 and 6 lack statistical analysis. If the data representation does not allow the depiction of these analyses, the investigators should include it as supplementary information

Author Response

Thank you for taking the time to review our revised manuscript, and to express your improved opinion about our work. Special thanks for the useful new suggestions how to further develop the article.

The new concerns are addressed as follows.

  1. The conclusions are still not fully supported by the data. The authors need to demonstrate that indeed caspase-dependent, ER-dependent and JNK-dependent cell death is triggered by Palmitate. This requires straightforward experiments using pharmacological regulators of these cascades.

We agree that our study would be unsubstantiated without a thoroughly definition of palmitate toxicity and its molecular mechanism, with special regards on the role of the ER stress. However, the fact that palmitate damages or even destroys a wide variety of cells, particularly pancreatic β-cells and insulinoma cells through caspase-dependent, ER-dependent and JNK-dependent mechanisms is not the conclusion but the well-established background of our study. There is a plethora of experimental evidence for the role of ER stress and the UPR signaling pathways including XBP-1 mRNA cleavage, phosphorylation of eukaryotic initiation factor 2α (eIF2α), activation of stress-activated protein kinase (SAPK) / c-Jun N-terminal kinase (JNK) and diverse caspase cascades, and a few of the most relevant papers in this field were cited in the original version of the manuscript. We thank the reviewer for highlighting that some specific details of these mechanisms might need further emphasis in the Introduction of our manuscript. The first paragraph has been extended (lines 43-47) and a comprehensive review article [Biden, T. J.; Boslem, E.; Chu, K.Y; Sue, N. Lipotoxic endoplasmic reticulum stress, β cell failure, and type 2 diabetes mellitus. Trends Endocrinol Metab 2014, 25, 389-398, doi: 10.1016/j.tem.2014.02.003] has been added to the list of references for a more appropriate demonstration of the scientific background of our study.

  1. Figure 5 and 6 lack statistical analysis. If the data representation does not allow the depiction of these analyses, the investigators should include it as supplementary information

The statistical analysis was missing because the three dimensional diagram indeed did not allow their depiction. Nevertheless, we agree that these data should be presented, and therefore, according to the reviewer’s suggestion, these results and their analyses have been arranged in 4 supplementary tables, which are now supplied with the article.

References

  1. Simon-Szabó L, Kokas M, Mandl J, Kéri G, Csala M. Metformin attenuates palmitate-induced endoplasmic reticulum stress, serine phosphorylation of IRS-1 and apoptosis in rat insulinoma cells. PLoS One. 2014;9(6):e97868. Epub 2014/06/04. doi: 10.1371/journal.pone.0097868. PubMed PMID: 24896641; PubMed Central PMCID: PMCPMC4045581.
  2. Shimabukuro M, Zhou YT, Levi M, Unger RH. Fatty acid-induced beta cell apoptosis: a link between obesity and diabetes. Proc Natl Acad Sci U S A. 1998;95(5):2498-502. doi: 10.1073/pnas.95.5.2498. PubMed PMID: 9482914; PubMed Central PMCID: PMCPMC19389.
  3. Cnop M, Igoillo-Esteve M, Cunha DA, Ladrière L, Eizirik DL. An update on lipotoxic endoplasmic reticulum stress in pancreatic beta-cells. Biochem Soc Trans. 2008;36(Pt 5):909-15. doi: 10.1042/BST0360909. PubMed PMID: 18793160.
  4. Laybutt DR, Preston AM, Akerfeldt MC, Kench JG, Busch AK, Biankin AV, et al. Endoplasmic reticulum stress contributes to beta cell apoptosis in type 2 diabetes. Diabetologia. 2007;50(4):752-63. Epub 2007/02/01. doi: 10.1007/s00125-006-0590-z. PubMed PMID: 17268797.
  5. Karaskov E, Scott C, Zhang L, Teodoro T, Ravazzola M, Volchuk A. Chronic palmitate but not oleate exposure induces endoplasmic reticulum stress, which may contribute to INS-1 pancreatic beta-cell apoptosis. Endocrinology. 2006;147(7):3398-407. Epub 2006/04/06. doi: 10.1210/en.2005-1494. PubMed PMID: 16601139.
  6. Cunha DA, Hekerman P, Ladrière L, Bazarra-Castro A, Ortis F, Wakeham MC, et al. Initiation and execution of lipotoxic ER stress in pancreatic beta-cells. J Cell Sci. 2008;121(Pt 14):2308-18. Epub 2008/06/17. doi: 10.1242/jcs.026062. PubMed PMID: 18559892; PubMed Central PMCID: PMCPMC3675788.
  7. Choi SE, Lee YJ, Jang HJ, Lee KW, Kim YS, Jun HS, et al. A chemical chaperone 4-PBA ameliorates palmitate-induced inhibition of glucose-stimulated insulin secretion (GSIS). Arch Biochem Biophys. 2008;475(2):109-14. Epub 2008/04/18. doi: 10.1016/j.abb.2008.04.015. PubMed PMID: 18455496.
  8. Kwak HJ, Yang D, Hwang Y, Jun HS, Cheon HG. Baicalein protects rat insulinoma INS-1 cells from palmitate-induced lipotoxicity by inducing HO-1. PLoS One. 2017;12(4):e0176432. Epub 2017/04/26. doi: 10.1371/journal.pone.0176432. PubMed PMID: 28445528; PubMed Central PMCID: PMCPMC5405981.
  9. El-Assaad W, Buteau J, Peyot ML, Nolan C, Roduit R, Hardy S, et al. Saturated fatty acids synergize with elevated glucose to cause pancreatic beta-cell death. Endocrinology. 2003;144(9):4154-63. doi: 10.1210/en.2003-0410. PubMed PMID: 12933690.
  10. Chen YY, Sun LQ, Wang BA, Zou XM, Mu YM, Lu JM. Palmitate induces autophagy in pancreatic β-cells via endoplasmic reticulum stress and its downstream JNK pathway. Int J Mol Med. 2013;32(6):1401-6. Epub 2013/10/18. doi: 10.3892/ijmm.2013.1530. PubMed PMID: 24142192.
  11. Liu X, Zeng X, Chen X, Luo R, Li L, Wang C, et al. Oleic acid protects insulin-secreting INS-1E cells against palmitic acid-induced lipotoxicity along with an amelioration of ER stress. Endocrine. 2019;64(3):512-24. Epub 2019/02/18. doi: 10.1007/s12020-019-01867-3. PubMed PMID: 30778898.
  12. Sommerweiss D, Gorski T, Richter S, Garten A, Kiess W. Oleate rescues INS-1E β-cells from palmitate-induced apoptosis by preventing activation of the unfolded protein response. Biochem Biophys Res Commun. 2013;441(4):770-6. Epub 2013/11/01. doi: 10.1016/j.bbrc.2013.10.130. PubMed PMID: 24189472.
  13. Sargsyan E, Artemenko K, Manukyan L, Bergquist J, Bergsten P. Oleate protects beta-cells from the toxic effect of palmitate by activating pro-survival pathways of the ER stress response. Biochim Biophys Acta. 2016;1861(9 Pt A):1151-60. Epub 2016/06/22. doi: 10.1016/j.bbalip.2016.06.012. PubMed PMID: 27344025.
  14. Gehrmann W, Würdemann W, Plötz T, Jörns A, Lenzen S, Elsner M. Antagonism Between Saturated and Unsaturated Fatty Acids in ROS Mediated Lipotoxicity in Rat Insulin-Producing Cells. Cell Physiol Biochem. 2015;36(3):852-65. Epub 2015/05/27. doi: 10.1159/000430261. PubMed PMID: 26044490.
  15. Zhu Q, Zhong JJ, Jin JF, Yin XM, Miao H. Tauroursodeoxycholate, a chemical chaperone, prevents palmitate-induced apoptosis in pancreatic β-cells by reducing ER stress. Exp Clin Endocrinol Diabetes. 2013;121(1):43-7. Epub 2012/09/12. doi: 10.1055/s-0032-1321787. PubMed PMID: 22972029.
  16. Green CD, Olson LK. Modulation of palmitate-induced endoplasmic reticulum stress and apoptosis in pancreatic β-cells by stearoyl-CoA desaturase and Elovl6. Am J Physiol Endocrinol Metab. 2011;300(4):E640-9. Epub 2011/01/25. doi: 10.1152/ajpendo.00544.2010. PubMed PMID: 21266672.
  17. Huang S, Zhu M, Wu W, Rashid A, Liang Y, Hou L, et al. Valproate pretreatment protects pancreatic β-cells from palmitate-induced ER stress and apoptosis by inhibiting glycogen synthase kinase-3β. J Biomed Sci. 2014;21:38. Epub 2014/05/04. doi: 10.1186/1423-0127-21-38. PubMed PMID: 24884462; PubMed Central PMCID: PMCPMC4084580.
  18. Lai E, Bikopoulos G, Wheeler MB, Rozakis-Adcock M, Volchuk A. Differential activation of ER stress and apoptosis in response to chronically elevated free fatty acids in pancreatic beta-cells. Am J Physiol Endocrinol Metab. 2008;294(3):E540-50. Epub 2008/01/15. doi: 10.1152/ajpendo.00478.2007. PubMed PMID: 18198352.
  19. Pirot P, Ortis F, Cnop M, Ma Y, Hendershot LM, Eizirik DL, et al. Transcriptional regulation of the endoplasmic reticulum stress gene chop in pancreatic insulin-producing cells. Diabetes. 2007;56(4):1069-77. doi: 10.2337/db06-1253. PubMed PMID: 17395747.
  20. Cnop M, Ladriere L, Hekerman P, Ortis F, Cardozo AK, Dogusan Z, et al. Selective inhibition of eukaryotic translation initiation factor 2 alpha dephosphorylation potentiates fatty acid-induced endoplasmic reticulum stress and causes pancreatic beta-cell dysfunction and apoptosis. J Biol Chem. 2007;282(6):3989-97. Epub 2006/12/08. doi: 10.1074/jbc.M607627200. PubMed PMID: 17158450.
  21. Wang W, Zhang D, Zhao H, Chen Y, Liu Y, Cao C, et al. Ghrelin inhibits cell apoptosis induced by lipotoxicity in pancreatic beta-cell line. Regul Pept. 2010;161(1-3):43-50. Epub 2010/01/14. doi: 10.1016/j.regpep.2009.12.017. PubMed PMID: 20079380.
  22. Lupi R, Dotta F, Marselli L, Del Guerra S, Masini M, Santangelo C, et al. Prolonged exposure to free fatty acids has cytostatic and pro-apoptotic effects on human pancreatic islets: evidence that beta-cell death is caspase mediated, partially dependent on ceramide pathway, and Bcl-2 regulated. Diabetes. 2002;51(5):1437-42. doi: 10.2337/diabetes.51.5.1437. PubMed PMID: 11978640.
  23. Solinas G, Naugler W, Galimi F, Lee MS, Karin M. Saturated fatty acids inhibit induction of insulin gene transcription by JNK-mediated phosphorylation of insulin-receptor substrates. Proc Natl Acad Sci U S A. 2006;103(44):16454-9. Epub 2006/10/18. doi: 10.1073/pnas.0607626103. PubMed PMID: 17050683; PubMed Central PMCID: PMCPMC1637603.
  24. Martinez SC, Tanabe K, Cras-Méneur C, Abumrad NA, Bernal-Mizrachi E, Permutt MA. Inhibition of Foxo1 protects pancreatic islet beta-cells against fatty acid and endoplasmic reticulum stress-induced apoptosis. Diabetes. 2008;57(4):846-59. Epub 2008/01/03. doi: 10.2337/db07-0595. PubMed PMID: 18174526.
  25. Choi SE, Lee SM, Lee YJ, Li LJ, Lee SJ, Lee JH, et al. Protective role of autophagy in palmitate-induced INS-1 beta-cell death. Endocrinology. 2009;150(1):126-34. Epub 2008/09/04. doi: 10.1210/en.2008-0483. PubMed PMID: 18772242.
  26. Komiya K, Uchida T, Ueno T, Koike M, Abe H, Hirose T, et al. Free fatty acids stimulate autophagy in pancreatic β-cells via JNK pathway. Biochem Biophys Res Commun. 2010;401(4):561-7. Epub 2010/10/01. doi: 10.1016/j.bbrc.2010.09.101. PubMed PMID: 20888798.
  27. Cnop M, Hannaert JC, Hoorens A, Eizirik DL, Pipeleers DG. Inverse relationship between cytotoxicity of free fatty acids in pancreatic islet cells and cellular triglyceride accumulation. Diabetes. 2001;50(8):1771-7. doi: 10.2337/diabetes.50.8.1771. PubMed PMID: 11473037.
  28. Cnop M, Ladrière L, Igoillo-Esteve M, Moura RF, Cunha DA. Causes and cures for endoplasmic reticulum stress in lipotoxic β-cell dysfunction. Diabetes Obes Metab. 2010;12 Suppl 2:76-82. doi: 10.1111/j.1463-1326.2010.01279.x. PubMed PMID: 21029303.
  29. Back SH, Kang SW, Han J, Chung HT. Endoplasmic reticulum stress in the β-cell pathogenesis of type 2 diabetes. Exp Diabetes Res. 2012;2012:618396. Epub 2011/09/08. doi: 10.1155/2012/618396. PubMed PMID: 21915177; PubMed Central PMCID: PMCPMC3170700.
  30. Volchuk A, Ron D. The endoplasmic reticulum stress response in the pancreatic β-cell. Diabetes Obes Metab. 2010;12 Suppl 2:48-57. doi: 10.1111/j.1463-1326.2010.01271.x. PubMed PMID: 21029300.
  31. Allagnat F, Cunha D, Moore F, Vanderwinden JM, Eizirik DL, Cardozo AK. Mcl-1 downregulation by pro-inflammatory cytokines and palmitate is an early event contributing to β-cell apoptosis. Cell Death Differ. 2011;18(2):328-37. Epub 2010/08/27. doi: 10.1038/cdd.2010.105. PubMed PMID: 20798690; PubMed Central PMCID: PMCPMC3131897.
  32. Natalicchio A, Labarbuta R, Tortosa F, Biondi G, Marrano N, Peschechera A, et al. Exendin-4 protects pancreatic beta cells from palmitate-induced apoptosis by interfering with GPR40 and the MKK4/7 stress kinase signalling pathway. Diabetologia. 2013;56(11):2456-66. Epub 2013/08/31. doi: 10.1007/s00125-013-3028-4. PubMed PMID: 23995397.
  33. Biden TJ, Boslem E, Chu KY, Sue N. Lipotoxic endoplasmic reticulum stress, β cell failure, and type 2 diabetes mellitus. Trends Endocrinol Metab. 2014;25(8):389-98. Epub 2014/03/18. doi: 10.1016/j.tem.2014.02.003. PubMed PMID: 24656915.

Round 3

Reviewer 2 Report

The investigators refused to address the concerns from the reviewer. They need to demonstrate that indeed caspase-dependent, ER-dependent and JNK-dependent cell death is triggered by Palmitate. While previous studies have demonstrated a role for these pathways, they have been done either in different experimental models or conditions. 

Author Response

Thank you for reviewing our manuscript again. We are sorry we could not convince you that our study is based on a substantial scientific background.